# Characteristic Profile of the Hazardous, Nutritional, and Taste-Contributing Compounds during the Growth of *Argopecten irradians* with Different Shell Colors

**DOI:** 10.3390/foods12234354

**Published:** 2023-12-02

**Authors:** Teng Wang, Jixing Peng, Xinnan Zhao, Yichen Lin, Dongru Song, Yanfang Zhao, Yanhua Jiang, Haiyan Wu, Qianqian Geng, Guanchao Zheng, Mengmeng Guo, Zhijun Tan

**Affiliations:** 1Key Laboratory of Testing and Evaluation for Aquatic Product Safety and Quality, Ministry of Agriculture and Rural Affairs, Yellow Sea Fisheries Research Institute, Chinese Academy of Fishery Sciences, Qingdao 266071, China; wangteng20210227@163.com (T.W.); zhaoxn@ysfri.ac.cn (X.Z.); linyc@ysfri.ac.cn (Y.L.); drsong97@163.com (D.S.); zhaoyf@ysfri.ac.cn (Y.Z.); jiangyh@ysfri.ac.cn (Y.J.); wuhy@ysfri.ac.cn (H.W.); gengqq@ysfri.ac.cn (Q.G.); zhenggc@ysfri.ac.cn (G.Z.); guomm@ysfri.ac.cn (M.G.); tanzj@ysfri.ac.cn (Z.T.); 2State Key Laboratory of Mariculture Biobreeding and Sustainable Goods, Yellow Sea Fisheries Research Institute, Chinese Academy of Fishery Sciences, Qingdao 266071, China

**Keywords:** bay scallop, shell color, growth, nutrient composition, hazard

## Abstract

Bay scallops (*Argopecten irradians; A. irradians*) are shellfish with high nutritional and economic value. However, nutritional studies on *A. irradians* with different shell colors are limited. This study examines the hazardous, nutritional, and taste-contributing compounds during the growth of *A. irradians* with different shell colors. During the growth of *A. irradians*, the hazardous contents were below the standard limit. Changes in the nutritional and taste-contributing compounds between months were more significant than shell color. Bay scallops had more fats, total fatty acids, and taste-contributing compounds in August and more proteins, essential fatty acids, vitamin D, vitamin B_12_, Cu, and Zn in September and October. In October, the golden shell color strain had more proteins, essential fatty acids, vitamin D, vitamin B_12_, Cu, and Zn, while the purple shell color strain had more taste-contributing compounds. *A. irradians* had better taste in August and higher nutritional value in September and October. In October, the golden shell color strain has higher nutritional value, and the purple shell color strain has better commercial value and taste. The correlation analysis indicates that the nutritional quality of bay scallops is affected by age (months), shell color, and seawater environment.

## 1. Introduction

Scallops are important marine shellfish in China, according to the recent China Fishery Statistics Yearbook [1]. In 2022, the total yield of scallops in China reached 1.792 million tons, positioning scallops as the third major farmed marine commercial shellfish, following oysters and clams. Shandong is the largest production area for scallops; it yields about 983,900 tons. The reports reveal four common economic species: *Chlamys farreri*, *Mizuhopecten yessoensis*, *Mimachlamys nobilis*, and *Argopecten irradians* (*A. irradians*). While *A. irradians* is native to the Atlantic US, it was introduced to China in 1982 following breakthroughs in aquaculture technology during the 1990s [2]. *A. irradians* has become a major culture scallop in China; it is placed in cages in late July, cultured during August and September, and harvested and sold in October [3,4]. *A. irradians* is a valuable food in terms of being rich in protein and poor in fat [3], as well as having many bioactive compounds, including taurine, choline, polysaccharides, eicosapentaenoic acid (EPA), docosahexaenoic acid (DHA), and all essential amino acids [5].

Previously limited research on the nutritional composition of *A. irradians* focused on the comparative assessment of nutritional excellence among different varieties of scallops and found that *A. irradians* is superior to *Mizuhopecten yessoensis* in terms of both nutritional value and economics [6]. More research focused on bay scallops and has found that bay scallops from different origins in northern China have differences in proximate composition. Different shell color bay scallops also differ in growth and survival rates during July and August [7,8]. Accordingly, evaluating the quality and safety profile during the growth of *A. irradians* with different colors is necessary. This study aims to comprehensively analyze seawater environmental parameters, microbial and chemical hazards, biometric measurements, nutritional components, and taste-contributing compounds in cultured *A. irradians* during its growth in Rongcheng Ailian Bay. We also provide valuable insights applicable to *A. irradians* aquaculture, production methodologies, and innovative processing techniques. Ultimately, the research provides a valuable reference for the culture of either golden or purple shell scallops and contributes to the market promotion of golden shell scallops.

## 2. Materials and Methods

### 2.1. Reagents and Instruments 

Chloroform, methanol, hexane, isopropyl alcohol, acetonitrile, acetone, ammonium formate, and formic acid (chromatographically pure) were obtained from Merck & Co Inc (Rahway, NJ, USA). Hydrochloric acid, sodium hydroxide, boron trifluoride, sodium chloride, sulfuric acid, and magnesium chloride (chemically pure) were obtained from Sinopharm Chemical Reagent Co. Ltd (Shanghai, China). Succinic acid, citric acid, malic acid standards, 20 amino acid standards, vitamin B_1_, vitamin B_3_, vitamin B_6_, vitamin B_12_, and vitamin E standards (purity ≥ 99%) were obtained from Dr. Ehrenstorfer(Augsburg, Germany). Tartaric acid standard (purity ≥ 99%) were obtained from Sigma-Aldrich. Nucleotide standards (six types) and betaine standard (purity ≥ 99%) were obtained from ANPEL(Shanghai, China). A total of 37 types of fatty acid methyl ester were mixed as standard (total 10 mg/mL) from ANPEL. Vitamin B_2_, vitamin B_5_, vitamin A, vitamin D_2_, and vitamin D_3_ standards, polychlorinated biphenyls mixed standard solution (first standard, purity ≥ 99%), and polyaromatic hydrocarbon mixed standard solution (purity ≥ 99%) were obtained from BePure(Beijing, China). Multi-element mixed standard solutions were obtained from the China National Center for Standard Materials(Beijing, China). A total of 10 types of paralytic shellfish toxin standards and three kinds of fat-soluble shellfish toxin standards were obtained from the National Research Council of Canada (Halifax, Nova Scotia, Canada).

The equipment included the following: an Agilent 7890B gas chromatograph (Agilent, Santa Clara, CA, USA), a Sciex 5500 Qtrap mass spectrometer with LC-20A liquid chromatograph (Sciex, Framingham, MA, USA; Shimadzu, Japan), an HWS-26 electric thermostatic water bath (Shanghai Shengke Instrument Equipment Co., Ltd.) (Shanghai, China), a KQ-300E ultrasonic extractor (Kunshan Ultrasonic Instrument Co., Ltd.) (Kunshan, China), an Himac CR 22GII high-speed refrigerated centrifuge (Hitachi, Tokyo, Japan), a Milli-Q ultrapure water meter (Millipore, MO, USA), a CHRIST Betal-8LDplus freeze dryer (CHRIST, Niedersachsen, Germany), a N-EVAP-24 nitrogen-blowing apparatus (Organomation, KS, USA), a Sartorius CPA 1003P Electronic Balance (Sartorius, Niedersachsen, Germany), an IKA VORTEX 2 Vortex Apparatus (IKA, Berlin, Germany), a MARS 6 Microwave Digestion apparatus (CEM, NC, USA), a Kjeltec2400 automatic nitrogen analyzer (FOSS, Hillerød, Denmark), an L-8800 automatic amino acid automatic analyzer (Hitachi, Tokyo, Japan), an Elan DRC II inductively coupled plasma emission spectrometer (PE, Boston, MA, USA), a Microplate Reader BIO-RAD 550 (Biorad, Hercules, CA, USA), and an HPLC Waters e2695 (Waters, Milford, MA, USA).

### 2.2. Sample Collection

Samples of seawater and two shell color strains of three-month-old *A. irradians* were collected each month from August 2021 to October 2021 in the Ailian Bay mariculture area of Weihai Changqing Marine Science and Technology Co., Ltd., in Rongcheng City, Shandong Province, China (Figure 1). Selected site measurements of water temperature, salinity, pH, dissolved oxygen, chlorophyll a, and nutrients (NH4-N, NO3-N, NO2-N, and PO4-P). For each month, from August to October, two different shell color samples were collected.

### 2.3. Sample Pretreatment

Four kg samples from each site were kept in ice and transported to the laboratory. Shell height, length, and width, as well as the total weight and tissue weights, were measured. All the samples were divided into two parts: one for determining microbial, chemical hazard, moisture, and taste-contributing compounds, and the other was freeze-dried and stored at −80 °C for subsequent fat, protein, hydrolyzed amino acid, fatty acid, vitamin, and mineral analysis.

### 2.4. Methods of Analysis

All of the methods listed here are available; the water temperature, salinity, pH, and dissolved oxygen were measured by a portable multiparameter meter (Multi3630IDS, WTW). The determination of chlorophyll-a was measured by spectrophotometry; the determination of the microbial and chemical hazard contents was carried out according to the National Food Safety Standards of China; the determination of nutritional composition (proximate composition, hydrolyzed amino acids, fatty acids, vitamins, and minerals) content was carried out according to the National Food Safety Standards of China; the taste-contributing compounds were measured by following the established methodology by Song et al. [9]. Details of the methods are available in the supplement. Statistical analysis was performed in triplicate (*n* = 3).

### 2.5. Data Analysis

All measurements were performed three times, from extraction to determination. All experimental data were expressed as mean ± standard deviation. The data were analyzed by one-way ANOVA using SPSS 20.0 software, and the data were compared using independent sample t-tests, variance chi-square tests using Leven’s test, and Duncan’s multiple comparisons were performed on the different data, with *p* < 0.05 as a significant difference. Spearman correlation analysis was performed, and Gephi (Version 0.9.2, Web Atlas, Paris, France) was used for the visual correlation network graphing, in which the size of the circles is related to the number of edges of the compounds, and the thickness of the lines is proportional to the spearman correlation value (|r|> 0.5, *p* < 0.05). Origin 2022 was used for data plotting.
Dressing percentage = (Soft tissue weight/Total weight × 100%)(1)
Amino acid scoring pattern (AAS) = mg of EAA in 1 g of protein of testsamples/ mg of EAA in 1 g of protein of (FAO/WHO) reference pattern × 100(2)

The amino acid ratio coefficient (RC) is calculated as
RC_k_ = AAS_k_/AAS(3)
(4)$$\mathrm{EAAI}=\sqrt[{n}]{({\mathrm{Leu}}^{a}/{\mathrm{Leu}}^{b})\;\times \;({\mathrm{Val}}^{a}/{\mathrm{Val}}^{b})\;\times \;\ldots \;\times \;({\mathrm{Lys}}^{a}/{\mathrm{Lys}}^{b})\;\times \;({\mathrm{His}}^{a}/{\mathrm{His}}^{b}})\;\times \;100$$

The amino acid ratio coefficient score (SRC) was calculated as
SRC = 100 − RSD (5)
(6)Atherogenicity index (AI)=(C12:0+4×C14:0+C16:0)/(ΣMUFAs+ΣPUFAs)
(7)$$\mathrm{Thrombosis}\;\mathrm{index}\;(\mathrm{TI})\;=\frac{\left(C14:0+C16:0+C18:0\right)}{\{0.5\times \Sigma \mathrm{MUFAs}+0.5\times \Sigma n6\mathrm{PUFAs}+3\times \Sigma n3\mathrm{PUFAs}+(n3/n6)\}}$$
(INQ, index of nutrition quality) = (consumed amount of a nutrient per 1000 kcal/recommended dietary allowance or adequate intake of that nutrient per 1000 kcal)(8)

Note: Leu^a^, Val^a^, …, His^a^ represents the essential amino acid content of the sample; Leu^b^, Val^b^, …, His^b^ is the standardized score of essential amino acids, and n is the number of essential amino acids; AAS_k_ is the amino acid ratio of the K essential amino acid in the sample; AAS is the average value of essential amino acid ratio in the sample; RSD is the variation number of RC.

## 3. Results and Discussion 

### 3.1. Seawater Environment 

Appendix A shows that the environmental parameters varied from August to October. The water temperature and salinity ranged from 17.1 °C (October) to 23.5 °C (September) and from 23.20 (September) to 26.70 (August), respectively. The dissolved oxygen content in the aquaculture waters was 6.31–9.79 mg/L. Surface water acidity was relatively stable around pH 8 despite some fluctuations in the culturing area. The chlorophyll-a content of these waters was 1.71–3.78 mg/m^3^. Four nutrients (NH_4_-N, NO_3_-N, NO_2_-N, and PO_4_-P) were detected in the aquaculture waters, and the dissolved inorganic nitrogen content was 0.12–1.31 mg/L, the phosphate content was 0.02–0.10 mg/L, and the N/P ratio was 5.06–17.05. These environmental parameters of seawater are suitable for *A. irradians* growth, according to previous research [10,11].

### 3.2. Biometric Measurements 

Table 1 summarizes the biometric measurements of *A. irradians* during growth. The shell length of the two shell color bay scallops increased from 35.2 to 68.3 mm (golden shell, 35.2–67.2 mm; purple shell, 38.3–68.3 mm), the total weight increased from 7.64 to 29.26 g (golden shell, 7.64–29.26 g; purple shell, 8.22–28.75 g), and the dressing percentage was 43.83–50.64% (golden shell, 43.83–48.95%; purple shell, 45.98–50.64%). In October, the two shell color bay scallops had insignificant differences (*p* > 0.05) in total weight, soft tissue weight, and dressing percentage.

### 3.3. Microbial and Chemical Hazards 

Appendix A and Figure 2 reveal the microbial and chemical hazard results, showing that *Escherchia coli* ranged from 0.62 to 2.06 MPN/g in the golden and purple shell color strains, which meets the standard of the second production area (<4.60 MPN/g); *Vibrio parahaemolyticus* ranged from 7.20 to 23.00 MPN/g, which meets the criteria for the pathogenic bacteria limit in food (<100 MPN/g).

Four harmful heavy metals were detected: lead (Pb) (0.10–0.28 vs. 1.5 mg/kg), cadmium (Cd) (0.35–0.66 vs. 1.5 mg/kg), arsenic (As) (0.24–0.35 vs. 0.5 mg/kg), and mercury (Hg) (10.31–16.06 µg/kg vs. 0.5 mg/kg), which were lower than the standard limits [12]. Shellfish toxins were detected as being paralytic and fat-soluble, and neither was detected in the two different shell color *A. irradians*. A total of seven polychlorinated biphenyls (PCBs) (Appendix A) and 16 polycyclic aromatic hydrocarbons (PAHs) (Appendix A) were detected in the organic pollutants: four PCBs were detected with a total content range of 0.91–29.22 µg/kg, including PCB52, PCB101, PCB118, and PCB180, much lower than the standard limit value (500 µg/kg) [13]; a total of 16 PAHs is lower than the limit of qualification [13]. The microbial and chemical hazards in the bay scallop samples of the two shell colors were under the safety limits and were safe for consumption. 

### 3.4. Proximate Composition 

Figure 3 and Appendix A show the proximate composition of *A. irradians* during growth; the protein and fat contents were 56.10–64.39% (golden shell, 58.05–64.39%; purple shell, 56.10–57.41%) and 4.48–11.03% (golden shell, 4.77–9.56%; purple shell, 4.48–11.03%) (dry samples), respectively, which is consistent with Wang Hong [14]. 

Protein is the major component in the samples, and the protein content determines the nutritional value and popularity of food products [15]. From August to October, *A. irradians* protein content increased significantly (*p* < 0.05); this indicates that most of the energy reserves were stored in *A. irradians*, which prepares for growth and gametogenesis [16] and was higher in the golden shell color strain than in the purple shell color strain. In the harvest period, the golden shell color strain had a higher protein content (*p* < 0.05). Fats are the main structural substances of cells and important energy reserves of the organism [17]. The total fat content of the golden and purple shell color strains was significantly higher in August than in September and October (*p* < 0.05). This may be because the bay scallop is in spawning season [18]. 

### 3.5. Hydrolyzed Amino Acid

Seafoods containing high-quality protein are valuable sources of nutrition in the daily human diet [19,20,21]. The content, type, and composition of essential amino acids determine the nutritional value of the protein in food. The closer the composition ratio is to that of essential amino acids in the human body, the higher the quality and practical value of the protein. The RC (ratio coefficient of amino acid), SRC (score of RC), and EAAI (essential amino acid index) values of essential amino acids [22] of the two shell color scallops were analyzed from August to October for a nutritional evaluation (Table 2).

Appendix A shows the contents of 16 hydrolyzed amino acids detected in *A. irradians*: eight essential and eight non-essential amino acids, with a range of 42.10–56.99 g/100 g from August to October (golden shell, 42.90–56.99 g/100 g; purple shell, 42.10–47.45 g/100 g). During its growth, the golden and purple shell color strains had the highest total amino acid content in September and October, respectively (*p* < 0.05). In the harvest period, the two shell color bay scallops had insignificant differences (*p* > 0.05) in the total amino acids.

Essential amino acids are an important index for assessing food protein nutrition quality, and whether the eight essential amino acids needed by humans match human dietary proteins is an important index for protein quality. The essential amino acid content in *A. irradians* growth was 11.31–12.81 g/100 g (golden shell, 12.19–12.81 g/100 g; purple shell, 11.31–12.10 g/100 g), with an insignificant difference (*p* > 0.05) between the two shell color bay scallops in the harvest period. The essential amino acid index (EAAI)—a common index evaluating the nutritional value of food—indicates the proximity of the essential amino acids in the sample and the standard protein (ovalbumin); the higher the EAAI, the higher the nutritional value of the protein. 

The EAAI of *A. irradians* during growth was 44.24–48.07 (golden shell, 44.24–47.05; purple shell, 44.86–48.07). The RC value can provide a reference for rationalizing the diet, revealing that isoleucine, phenylalanine, and tyrosine in *A. irradians* conformed to the ideal pattern (RC ≈ 1); lysine and threonine were in excess (RC > 1), and leucine, methionine, and valine were deficient (RC < 1). Therefore, bay scallops can complement cereals to supplement the lysine that is deficient in cereals. The SRC values were 92.33–98.21 (golden shell, 92.33–95.83; purple shell, 92.66–98.21), indicating that the essential amino acid composition of *A. irradians* during growth was reasonable.

### 3.6. Fatty Acid Content 

Appendix A and Figure 4 display changes in fatty acid contents and composition from August to October in *A. irradians*; a total of 14 fatty acids were determined in *A. irradians*: five saturated fatty acids (SFAs), three monounsaturated fatty acids (MUFAs), and six polyunsaturated fatty acids (PUFAs). 

The SFAs of *A. irradians* accounted for 26.33–34.94% (golden shell, 26.33–29.91%; purple shell, 29.25–34.94%) of the total fatty acids, most of which was palmitic acid (C16:0). MUFAs represented 16.65–21.92% (golden shell, 17.14–21.92%; purple shell, 16.65–20.59%), most of which was palmitoleic acid (C16:1n7), which is positively correlated to taste [23], and was significantly higher for the two shell color strains in August than in September and October (*p* < 0.05). This indicates that the two shell color strains had a better taste in August, with an insignificant difference (*p* > 0.05) in C16:1n7 content between the two different scallops in the harvest period.

The nutritional value of *A. irradians* was largely characterized by the composition of PUFAs, which accounted for 46.24–56.09% of the total fatty acids (golden shell, 48.17–56.09%; purple shell, 46.24–53.43%). EPA and DHA are important components of n-3 PUFAs in the bay scallops [24]. Existing studies show that if n-3/n-6 is unbalanced, it can lead to chronic diseases due to competition between n-3 and n-6 PUFAs for the same enzymes [25,26,27]. In this study, the ratios of n-3/n-6 of bay scallops were all greater than eight, suggesting that bay scallops can provide good n-3 unsaturated fatty acids and are suitable for human consumption. 

Their content was significantly higher (*p* < 0.05) in August and October than in September for the golden scallop, while significantly higher (*p* < 0.05) in August than in September and October for the purple scallop. Herein, the n-3/n-6 of *A. irradians* were all greater than one during growth, indicating that the *A. irradians* can provide n-3 unsaturated fatty acids suitable for human consumption. 

Atherogenicity (AI), the thrombogenicity index (TI), the levels of low and high-cholesterol fatty acids (HHs), and essential fatty acids (EFAs) have been used to assess the effects of fatty acid fractions in food on human health [28,29,30]. Appendix A reveals that bay scallops with different shell colors ranged from 0.37–0.60 for AI (golden shell, 0.37–0.51; purple shell, 0.47–0.60), 0.15–0.24 for TI (golden shell, 0.15–0.20; purple shell, 0.18–0.24), 1.75–3.05 for HHs (golden shell, 1.99–3.05; purple shell, 1.75–2.35), and 43.47–97.39 mg/100 g for EFAs (golden shell, 43.47–95.98 g/100 g; purple shell, 61.15–97.39 g/100 g). The fatty acid composition of *A. irradians* may offer health benefits for human diets.

In summary, these results revealed differences in the fatty acid composition of bay scallops with different shell colors during the growth and showed that the golden shell color strain had more EPA, DHA, and EFA contents and more fatty acid nutritional value in the harvest period.

### 3.7. Vitamin and Mineral Content

The vitamin content is an important dimension of the nutritional value of *A. irradians* since vitamins are essential for maintaining the normal physiological metabolism and health of the human body. This study determined 10 vitamins: A, D_2_, D_3_, E, B_1_, B_2_, B_3_, B_5_, B_6_, and B_12_, and the contents were consistent with previous investigations [31,32]. Water-soluble vitamins B_2_, B_3_, and B_5_ were rich, and their contents were 101.05–718.15 µg/100 g (golden shell, 101.05–490.64 g/100 g; purple shell, 168.36–718.15 g/100 g), 0.62–2.83 mg/100 g (golden shell, 0.62–2.83 g/100 g; purple shell, 1.05–2.49 g/100 g), and 139.25–744.07 µg/100 g (golden shell, 236.54–744.07 g/100 g; purple shell, 139.25–654.33 g/100 g), respectively. Fat-soluble vitamins A and E had high contents of 75.34–124.87 µg/100 g (golden shell, 75.34–124.87 g/100 g; purple shell, 77.88–105.72 g/100 g) and 0.93–3.05 mg/100 g (golden shell, 1.06–3.05 g/100 g; purple shell, 0.93–1.38 g/100 g), respectively (Appendix A and Figure 5). 

In order to evaluate the nutritional value of vitamins more accurately, the index of nutritional quality (INQ) was calculated by the recommended intake of nutrition and energy, which referred to the Chinese Food Composition Table [33] based on 2600 kal/d in 18–50-year-old males. Table 3 shows that the INQs of vitamin D and B_12_ were greater than 1, ranging from 5.16 to 17.82 (golden shell, 6.33–17.82; purple shell, 5.16–16.40) and 13.98 to 23.13 (golden shell, 13.98–23.13; purple shell, 16.98–21.66), respectively, indicating that these two vitamins are the dominant nutrients in *A. irradians*. Then, they can be used as an excellent dietary source of vitamin D and B_12_, which aligns with the report on *Chlamys farreri* [9].

Minerals cannot be synthesized in humans and, therefore, must be obtained through the diet [34,35,36]. Mineral elements include macro-elements and micro-elements; macro-elements are essential for human growth and development, whereas micro-elements participate in important metabolic activities and biochemical processes. We determined four macro-elements (K, Ca, Mg, and P) and five micro-elements (Fe, Cu, Zn, Mn, and Se) for the two shell color bay scallops (Appendix A). The INQ value of mineral elements of bay scallops was greater than 1, except for Se, indicating that bay scallops are good sources of mineral supplementation; among them, the INQ value of Cu and Zn is the highest. 

The Mn, Fe, and Cu contents from the golden shell color strain were the highest in September (*p* < 0.05), and the Se and Zn contents were the highest in October (*p* < 0.05). For the purple shell color strain, the Se contents were the highest in August (*p* < 0.05), the Mn and Fe contents reached the highest in September, and the Zn contents were the highest in October (*p* < 0.05). In the harvest period, the Fe, Zn, Se, and Cu contents were significantly higher in the golden shell color strain than in the purple (*p* < 0.05). Similar results can be found in oysters [37], which may be related to differences in growth metabolism across shell colors. 

### 3.8. Taste-Contributing Compounds and Taste Activity Value 

The unique taste of *A. irradians* is due to the flavor of amino acids, nucleotides, organic acids, and betaine. The TAV value can objectively and directly influence the overall taste of the food; TAV ≥ 1 indicates taste-active: the higher the TAV value, the more contribution to the taste sensation [38].

Table 4 and Appendix A show the content, composition, and TAV values of the free amino acids, nucleotides, organic acids, and betaine in *A. irradians* during its growth. Seventeen free amino acids, six nucleotides, four organic acids, and betaine were detected in the *A. irradians*. The total free amino acid (TFAA) content of *A. irradians* during its growth was 385.62–1132.86 mg/100 g (golden shell, 385.62–1132.86 mg/100 g; purple shell, 501.63–1068.44 g/100 g), and the two shell color strains had the highest free amino acid content in August (*p* < 0.05), which was higher in the golden shell color strain than in the purple shell strain (*p* < 0.05). Glycine and alanine impart a pleasant sweetness and freshness in aquatic products [39], and arginine imparts a pleasant odor to food products. Consequently, these three amino acids are the taste-active components of snow crabs, clams, and scallops [40]. The glycine content in *A. irradians* was 39.87–83.33 mg/100 g (golden shell, 46.37–72.10 mg/100 g; purple shell, 39.87–83.33 g/100 g), alanine was 49.52–116.19 mg/100 g (golden shell, 49.52–116.19 mg/100 g; purple shell, 51.43–110.32 g/100 g), and arginine was 10.33–31.40 mg/100 g (golden shell, 10.33–26.51 mg/100 g; purple shell, 10.42–31.40 g/100 g), which were significantly higher (*p* < 0.05) in August than in September and October in the golden shell color strain. The purple shell color strain had the highest alanine and glycine content in August (*p* < 0.05) and the highest arginine content in September (*p* < 0.05).

The glutamate content of the two shell color strains was 73.73–240.17 mg/100 g (golden shell, 73.73–236.71 mg/100 g; purple shell, 94.19–240.17 g/100 g), and the aspartic acid content was 24.29–130.62 mg/100 g (golden shell, 45.13–130.62 mg/100 g; purple shell, 24.26–98.82 g/100 g). Among the fresh amino acids, the TAV value of glutamic acid was greater than aspartic acid, whatever shell color or however many months old. Therefore, glutamic acid contributes the most to the taste attribute of *A. irradians*. In the bay scallop harvest period, the golden color had more glutamic acid (*p* < 0.05), while insignificant differences (*p* > 0.05) existed in the aspartic acid content between the two different bay scallops. Taurine can give a certain taste to food [41]; it had a content of 76.69–150.02 mg/100 g (golden shell, 76.69–150.02 mg/100 g; purple shell, 92.07–128.77 g/100 g) in *A. irradians*, with higher content in the purple shell color strain than the golden color strain in October (*p* < 0.05).

Taste-contributing nucleotides include adenine nucleotide (AMP), hypoxanthine nucleotide (IMP), cytosine nucleotide (CMP), guanine nucleotide (GMP), uracil nucleotide (UMP), and xanthine nucleotide (XMP). The highest TAV values for IMP and AMP in *A. irradians* indicated that IMP and AMP are the major taste-contributing nucleotides of *A. irradians*, ranging from 3.86 to 19.69 mg/100 g (golden shell, 3.86–19.69 mg/100 g; purple shell, 5.60–17.25 g/100 g) and 5.70 to 18.17 mg/100 g (golden shell, 5.70–11.63 mg/100 g; purple shell, 5.60–18.17 g/100 g), respectively. AMP is vital in the sweetness of *A. irradians*, providing a continuous fresh taste to *A. irradians* [42]. IMP and GMP are strong taste enhancers with a much stronger freshness than MSG. The EUC (equivalent umami concentration) values [43] of the two shell color scallops during growth (Figure 6) were the highest in August (*p* < 0.05); the EUC values of the golden and blue shell color strains were 8.08 and 6.80, respectively. The EUC value of the golden shell color strain in September and October was 1.08, and the EUC value of the purple shell color strain was 2.46 and 1.47, respectively. This demonstrates that the two shell color scallops had the best taste in August, with a better taste for the golden shell color strain than the purple shell strain, which had a better taste in September and October.

Among the four organic acids and betaine detected in *A. irradians*, succinic acid and betaine had TAV values greater than 1, whereas the other organic acids had much less than 1 regarding TAV values. Therefore, succinic acid is the main taste-contributing organic acid in bay scallops [44]. The golden shell color strain had the highest succinic acid and betaine contents in August (73.68 and 1266.48 mg/100 g, respectively; *p* < 0.05), whereas succinic acid had the lowest content in September (33.72 mg/100 g; *p* < 0.05), with an insignificant difference in betaine content between September and October (*p* > 0.05). The purple shell color strain had the highest succinic acid content of 48.14 mg/100 g in August, the highest betaine content of 722.08 mg/100 g in October, and the lowest succinic acid content and betaine content in September.

In summary, the taste of *A. irradians* was best in August when it was three months old, and the golden shell color strain was better than the purple shell color strain, while the taste of the purple shell color strain was better in October.

### 3.9. Correlation Analysis of Seawater Environment, Biometric Measurements, Nutrition Compounds, and Taste-Contributing Compounds in Bay Scallop

Herein, age in months, shell color, environmental factors, biometric measurements (shell length, width, height, dressing percentage, total weight, and soft tissue weight), nutritional components (protein, fat, total fatty acids, PUFAs, SFAs, EPA, DHA, total essential amino acids, and fat-soluble vitamins (A, D, and E), water-soluble vitamin B, minerals, and taste-contributing compounds (free amino acids, nucleotides, organic acid, and betaine) were analyzed by using correlation analysis (Figure 7). The size of the circles in the figure is related to the number of compound edges, and the thickness of the lines is proportional to the Spearman’s correlation value (|r| > 0.5, *p* < 0.05), with the orange and blue lines representing positive and negative correlations, respectively.

Age (months) was positively correlated with the biometric measurements and negatively correlated with fat, XMP, IMP, CMP, and tartaric. Shell color was correlated with protein. Temperature was positively correlated with Mn, dressing percentage, and UMP and negatively correlated with fat-soluble vitamins and Se. Dissolved oxygen and salinity were positively associated with the biometric measurements, protein, and Zn and negatively correlated with fat, VB, XMP, IMP, AMP, CMP, GMP, malic, and tartaric. Chlorophyll-a was positively correlated with Fat, VB, XMP, IMP, AMP, CMP, and GMP and negatively correlated with the biometric measurements, protein, and Zn.

In conclusion, the growth traits of *A. irradians* were affected by environmental factors, shell color, age (months), water temperature, salinity, dissolved oxygen, and chlorophyll-a. Accordingly, they are considered important factors in the growth environment, which is consistent with previous studies on *A. irradians* [45,46,47,48,49].

## 4. Conclusions

Herein, we comprehensively analyzed the microbial, chemically hazardous, nutritional, and taste-contributing compounds, as well as the biometric measurements, of two shell color strains of bay scallops from August to October. The golden shell color strain had a higher nutritional value, while the purple strain tasted better in October. This study elucidates the characteristic profiles regarding the nutritional quality and safety of two different shell color bay scallops during their growth. Golden et al. reported the role of aquatic foods in improving human nutrition, especially in pelagic fish and shellfish, which can supply critical nutrients such as calcium, iron, DHA+EPA, zinc, vitamin B_12_, and vitamin A [50]. This study can provide a reference to consumers and farmers about their choice of varieties, but the mechanisms underlying the differences in the nutritional quality of bay scallops with different shell color stains during their growth need further investigation.

## Figures and Tables

**Figure 1 foods-12-04354-f001:**
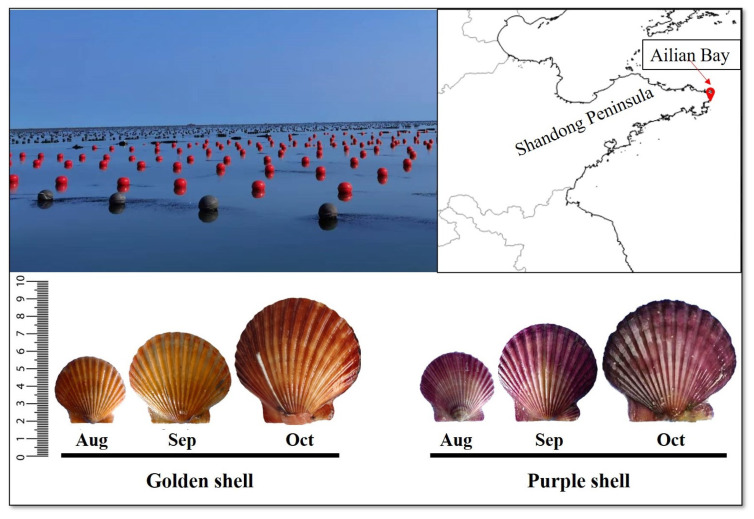
Sampling location and information.

**Figure 2 foods-12-04354-f002:**
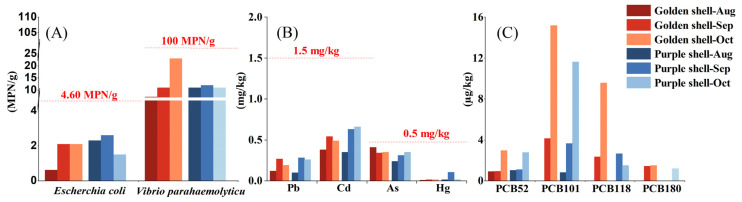
Analysis of microbial and chemical hazard components of shellfish during the growth of *A. irradians*. Pathogenic bacteria (**A**); heavy metals (**B**); PCBs (**C**).

**Figure 3 foods-12-04354-f003:**
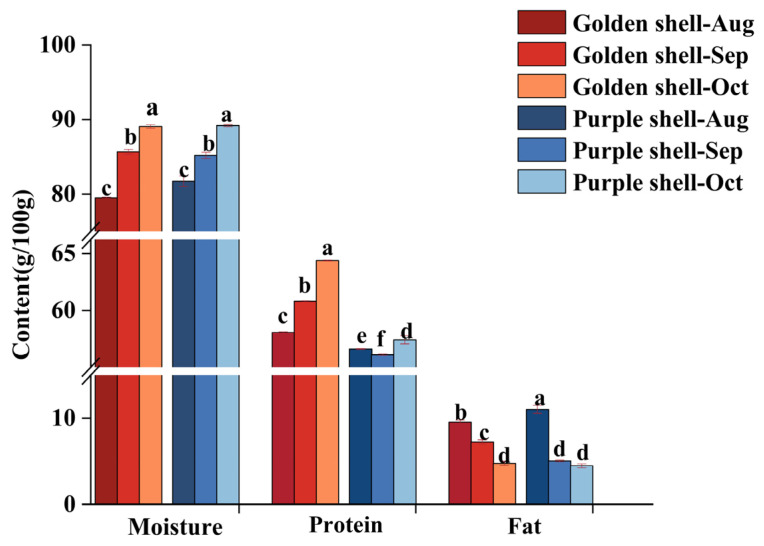
Changes in the proximate composition of *A. irradians.* The same letter means no significant difference between groups (*p* > 0.05), while values with different letters mean significant difference (*p* < 0.05) between groups.

**Figure 4 foods-12-04354-f004:**
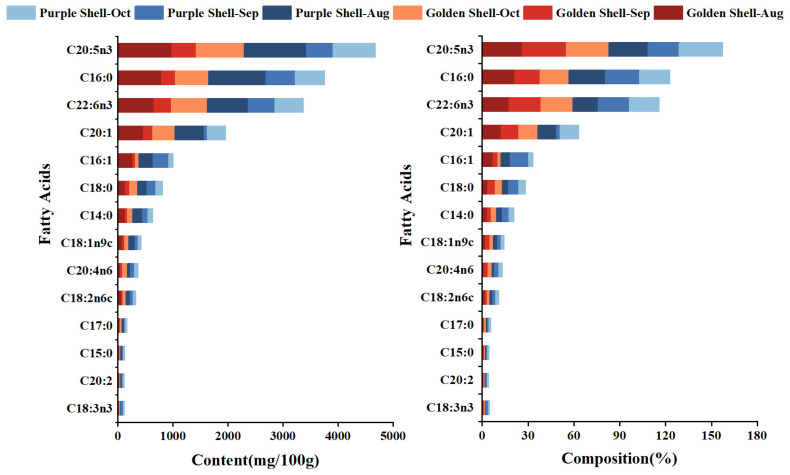
The fatty acid content and composition (% of total fatty acids) of *A. irradians* (dry samples).

**Figure 5 foods-12-04354-f005:**
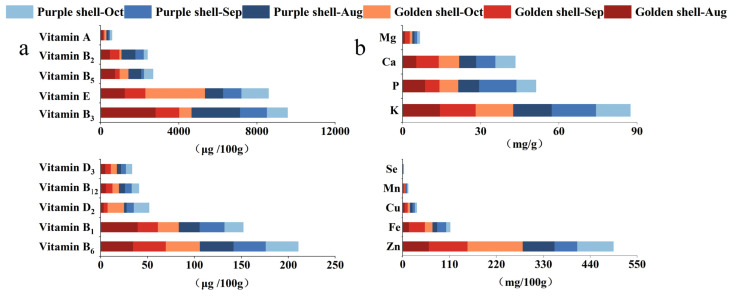
The vitamin (**a**) and mineral (**b**) content of *A. irradians* (dry samples).

**Figure 6 foods-12-04354-f006:**
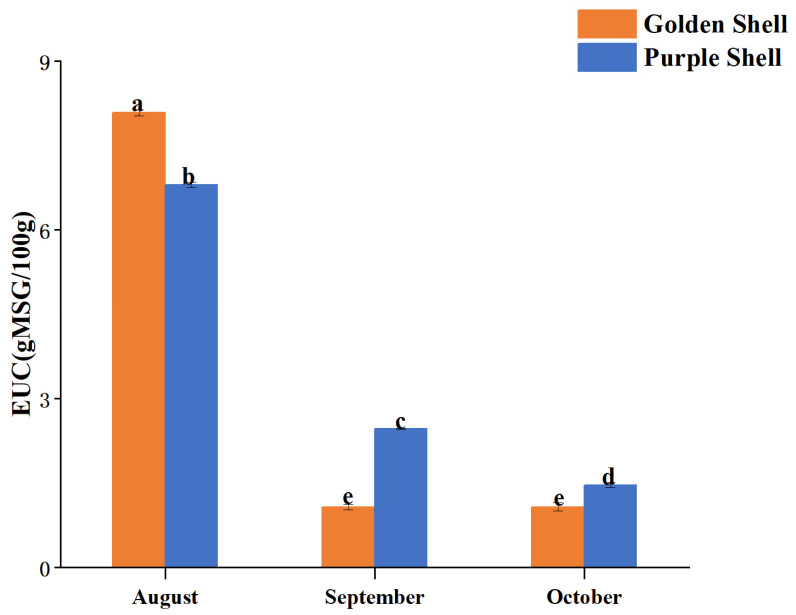
EUC of *A. irradians* during growth. The same letter means no significant difference between groups (*p* > 0.05), while values with different letters mean significant difference (*p* < 0.05) between groups.

**Figure 7 foods-12-04354-f007:**
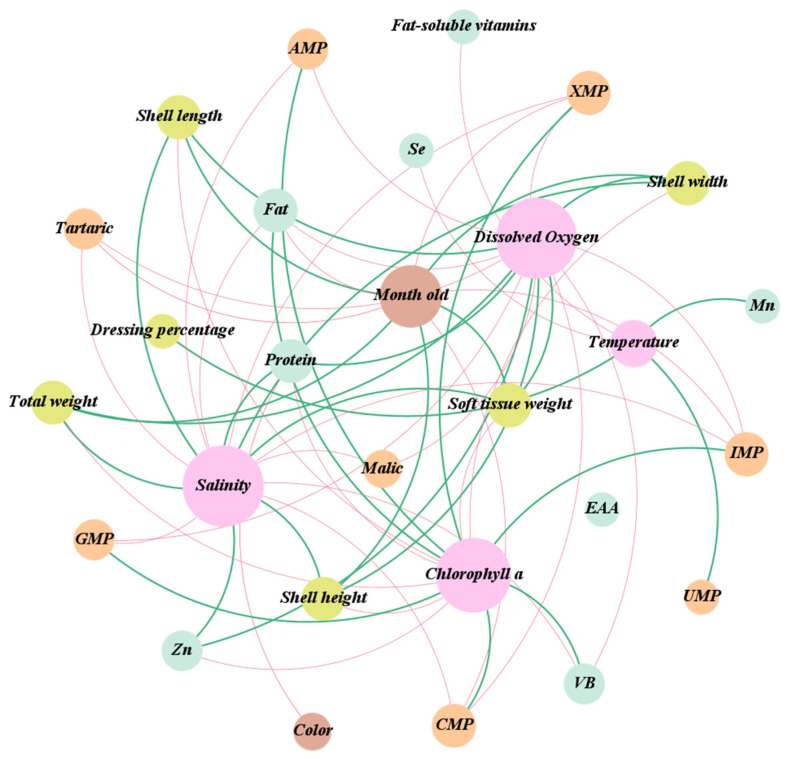
Correlation analysis of seawater environment, biometric measurements, nutrition compounds, and flavor substance in bay scallop.

**Table 1 foods-12-04354-t001:** Changes in the biometric measurements of *A. irradians*.

Index	Golden Shell			Purple Shell		
	August	September	October	August	September	October
Shell length (mm)	35.2 ± 0.2 ^f^	50.5 ± 0.6 ^d^	67.2 ± 0.2 ^b^	38.3 ± 0.3 ^e^	54.3 ± 0.3 ^c^	68.3 ± 0.3 ^a^
Shell height (mm)	39.2 ± 0.2 ^f^	51.3 ± 0.4 ^c^	62.5 ± 0.5 ^a^	40.2 ± 0.2 ^e^	50.4 ± 0.5 ^d^	60.7 ± 0.6 ^b^
Shell width (mm)	14.1 ± 0.1 ^d^	20.5 ± 0.5 ^b^	25.2 ± 0.2 ^a^	18.1 ± 0.6 ^c^	21.1 ± 0.2 ^ab^	21.4 ± 0.5 ^a^
Total weight (g)	7.64 ± 0.06 ^d^	17.44 ± 0.49 ^c^	29.26 ± 0.18 ^a^	8.22 ± 0.18 ^d^	19.06 ± 0.51 ^b^	28.75 ± 0.28 ^a^
Soft tissue weight (g)	3.52 ± 0.08 ^d^	7.64 ± 0.20 ^c^	14.32 ± 0.29 ^a^	3.92 ± 0.10 ^d^	8.76 ± 0.30 ^b^	14.56 ± 0.44 ^a^
Dressing percentage (%)	46.03 ± 0.68 ^bc^	43.83 ± 1.49 ^bc^	48.95 ± 1.28 ^ab^	47.73 ± 0.84 ^ab^	45.98 ± 2.58 ^bc^	50.64 ± 1.92 ^a^

Note: values are presented as means ± SD. Values in the same column that do not share the same superscript are significantly different (*p* < 0.05).

**Table 2 foods-12-04354-t002:** Changes in the essential amino acid coefficients and scores of *A. irradians* (dry samples).

RC	Golden Shell			Purple Shell		
	August	September	October	August	September	October
Isoleucine	1.02	1.06	0.92	0.93	1.01	1.00
Leucine	0.91	0.94	0.85	0.88	0.93	0.9
Lysine	1.49	1.43	1.29	1.49	1.48	1.42
Methionine	0.67	0.67	0.68	0.67	0.70	0.71
Phenylalanine + Tyrosine	1.00	0.98	0.97	1.11	0.99	1.01
Threonine	1.25	1.32	1.18	1.24	1.24	1.22
Valine	0.66	0.61	0.61	0.68	0.66	0.65
SRC (%)	95.81	95.83	92.33	98.21	92.66	95.32
EAAI	46.47	47.05	44.24	44.86	48.07	46.88

**Table 3 foods-12-04354-t003:** Changes in the nutritional quality indices of *A. irradians*.

Index	RNI	INQ					
		Golden Shell			Purple Shell		
		August	September	October	August	September	October
Vitamin A * (µg/100 g)	800	1.15	0.69	0.96	0.97	0.72	0.82
Vitamin D * (µg/100 g)	10	6.33	7.38	17.82	5.16	9.45	16.40
Vitamin E * (mg/100 g)	14 ^#^	0.65	0.56	1.60	0.49	0.49	0.72
Vitamin B_1_ * (mg/100 g)	1.4	0.21	0.11	0.12	0.12	0.14	0.1
Vitamin B_2_ * (mg/100 g)	1.4	2.54	2.58	0.53	3.77	2.26	0.88
Vitamin B_6_ * (mg/100 g)	1.4	0.18	0.18	0.19	0.19	0.18	0.18
Vitamin B_12_ * (µg/100 g)	2.4	18.20	23.13	13.98	21.27	21.66	16.98
Vitamin B_5_ * (mg/100 g)	5 ^#^	1.09	0.35	0.67	0.96	0.2	0.66
Vitamin B_3_ * (mg/100 g)	15	1.39	0.59	0.3	1.22	0.67	0.51
Ca (mg/100 g)	800	4.92	7.88	7.24	5.95	6.83	6.94
P (mg/100 g)	720	8.93	5.58	7.39	8.18	14.64	7.51
K (mg/100 g)	2000 ^#^	5.3	5.05	5.3	5.41	6.26	4.81
Mg (mg/100 g)	330	2.26	4.17	2.03	1.92	2.38	1.83
Fe (mg/100 g)	12	9.37	23.11	10.72	6.84	12.87	5.58
Zn (mg/100 g)	12.5	36.56	53.22	76.4	43.72	31.41	49.68
Se (µg/100 g)	60	0.03	0.02	0.07	0.08	0.05	0.05
Cu (mg/100 g)	0.80	48.90	65.62	51.93	53.57	50.19	34.96
Mn (mg/100 g)	4.5 ^#^	3.90	5.29	2.35	2.80	4.28	2.44

Note: reference intakes are calculated based on the recommended intakes or appropriate intakes for men, #: adequate intake, *: indicates that it was measured with dry samples.

**Table 4 foods-12-04354-t004:** Changes in the flavoring substances, threshold, and taste activity of *A. irradians*.

Compound	Taste Attribute	Golden Shell			Purple Shell			Taste Threshold (mg/100 mL)
		August	September	October	August	September	October	
Glutamic Acid	Umami (+)	7.89	2.46	3.39	8.01	3.52	3.14	30.00
Aspartic Acid	Umami (+)	1.31	0.55	0.45	0.99	0.24	0.45	100.00
Glycine	Sweet (+)	0.55	0.38	0.36	0.64	0.31	0.31	130.00
Alanine	Sweet (+)	1.94	0.83	0.86	1.84	1.20	0.86	60.00
Serine	Sweet (+)	0.30	0.09	0.11	0.33	0.14	0.16	150.00
* Threonine	Sweet (+)	0.11	0.02	0.04	0.11	0.06	0.05	260.00
Proline	Sweet/bitter (+)	0.45	0.13	0.19	0.44	0.06	0.25	300.00
Arginine	Bitter/Sweet (+)	0.53	0.21	0.21	0.48	1.37	0.21	50.00
Methionine *	Bitter/Sweet/sulfurous (−)	0.58	0.16	0.23	0.37	0.28	0.30	30.00
Histidine	Bitter (−)	1.88	0.18	0.62	1.71	0.25	0.85	20.00
* Leucine	Bitter (−)	0.07	0.09	0.04	0.08	0.05	0.05	190.00
* Isoleucine	Bitter (−)	0.12	0.06	0.05	0.13	0.13	0.07	90.00
* Phenylalanine	Bitter (−)	0.11	0.05	0.05	0.12	0.12	0.07	90.00
* Valine	Sweet/bitter (−)	0.27	0.12	0.10	0.28	0.26	0.14	40.00
* Lysine	Sweet/bitter (−)	0.25	0.12	0.08	0.27	0.20	0.13	50.00
IMP	Umami (+)	0.79	0.25	0.15	0.69	0.48	0.22	25.00
AMP	Umami/Sweet (+)	0.23	0.17	0.11	0.22	0.69	0.21	50.00
GMP	Umami (+)	0.14	0.09	0.09	0.10	0.09	0.13	12.5
Succinic	Sour/umami	6.95	3.18	4.60	4.54	1.41	4.60	10.6
Malic	Sour/bitter	0.43	0.24	0.21	0.39	0.12	0.26	49.6
Citric	Sour	0.56	0.44	0.53	0.39	0.04	0.45	45
Tartaric	Sour	0.53	0.42	0.36	0.48	0.50	0.33	1.5
Betaine	Sweet	5.07	1.76	1.79	2.28	1.60	3.55	250

Note: * indicates that it was essential amino acid.

## Data Availability

Data from the present study are available upon request from the corresponding author. The availability of the data is restricted to investigators based in academic institutions.

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
