# Peer review of "Characteristic Profile of the Hazardous, Nutritional, and Taste-Contributing Compounds during the Growth of Argopecten irradians with Different Shell Colors"

_foods, 2023, doi:10.3390/foods12234354_

Round 1
Reviewer 1 Report
Comments and Suggestions for Authors
Manuscript Foods-2707374 reports on the nutritional, chemical/microbiological hazards and taste contributing compounds of scallops of two different shell color during growth. This, in my opinion is a technically sound study that despite the fact that it does not propose anything new, it, however presents useful information on the value/quality of (A. irridians) scallops. The text should be proof-read by a native English speaker are there are numerous errors in the use of English.
My detailed comments follow the text sequence.
Title: change ‘ Taste Components ‘ to ‘taste contributing compounds’. Do the same throughout the text
l.42: change ‘rare’ to ‘limited’
l.44: rewrite in proper English
l.51: change ‘taste substance’ to ‘taste contributing compounds’
l.55: the authors should either delete this sentence or justify how this improvement is achieved.
l.56: change ‘nutrition’ to ‘nutritional content’
l.81: change ‘chromatography’ to ‘chromatograph’
l.93: mention the nutrient categories studied i.e. fatty acids, amino acids , etc.
l.103: change ‘Experimental method’ to ‘Methods of analysis’
l.110: if your variables were two (treatment/scallop shell color and time) the statistical treatment require two-way ANOVA
l.125-127: delete
l.141: define ‘dressing percentage’
l.157-158: I believe that the PCBs and PAHs determined should be at least mentioned in the text
l.187: give abbreviations RC, SRC, and EAAI in full
l.195: add ‘quality’ after ‘nutrition’
Table 2: statistical exponents are missing
l.211: add ‘content’ after ‘Fatty acid’
l.213: change ‘measured’ to ‘determined’
l.245: add ‘content’ after ‘Vitamin and mineral’
l.246: change ‘contents’ to ‘content’
l.255: change ‘performed’ to ‘calculated’
l.259-260: rewrite in proper English
l.263: delete ‘with’
l.263: sentence does not make sense. Rewrite in proper English
l.264: change ‘constant’ to ‘macro’
l.268: what o the authors mean by ‘unclear regular characteristics’ ?
l.269: change ‘is’ to ‘was’
Table 3: statistical exponents are missing
l.282: change ‘Taste substances’ to ‘taste contributing compounds’
l.304: change presentation’ to ‘attribute’
l.310, 326: change taste-presenting’ to ‘taste contributing’
l.31&: give abbreviation EUC in full
Table 4: statistical exponents are missing
l.342: delete ‘month-old’. Instead, add ‘monthly’ on l.347
l.359” change ‘are’ to ‘were’
l.368: change ‘taste substance’ to ‘taste contributing compounds’
l.370-371: rewrite in proper English
Based on the above, I recommend minor revision of the manuscript
Comments on the Quality of English Language
See my above comments.
Author Response
Thank you very much for taking the time to review this manuscript. Please find the detailed responses below and the corresponding revisions
1)Title: change “Taste Components” to “Taste contributing compounds”. Do the same throughout the text.
Reply: Done
2) l.42: change “rare” to “limited”
Reply: Done
3) l.44: rewrite in proper English
Reply: As suggested by the reviewer, we have rewritten l.44, the sentence has been rewritten to “and found that A. irradians is superior to Mizuhopecten yessoensis in terms of both nutrition and economics”.
4) l.51: change “taste substance” to “taste contributing compounds”
Reply: Done.
5) l.55: the authors should either delete this sentence or justify how this improvement is achieved.
Reply: As suggested by the reviewer, we have rewritten this sentence “contributes to the market promotion of golden shell scallop”.
6) l.56: change “nutrition” to “nutritional content”
Reply: Done.
7) l.81: change “chromatography” to “chromatograph”
Reply: Done.
8) l.93: mention the nutrient categories studied i.e. fatty acids, amino acids, etc.
Reply: “nutrients” indicates inorganic salt, and we have listed nutrients.
9) l.103: change “Experimental method” to “Methods of analysis”.
Reply: Done.
10) l.110: if your variables were two (treatment/scallop shell color and time) the statistical treatment require two-way ANOVA
Reply: Our research work was aimed to evaluate the nutritional quality of two shell-color Argopecten irradians during growth. Therefore, a one-way analysis of variance (ANOVA) was used. The above similar methods were also performed in the following references:
[a] Barrento, S., Marques, A., Teixeira, B., Anacleto, P., Vaz-Pires, P., Nunes, M. L. (2009). Effect of Season on the Chemical Composition and Nutritional Quality of the Edible Crab Cancer Pagurus. Journal of Agricultural and Food Chemistry, 57, 10814-10824.
[b] Barrento, S., Marques, A., Teixeira, B., Anacleto, P., Carvalho, M. L., Vaz-Pires, P., Nunes, M. L. (2009). Macro and trace elements in two populations of brown crab Cancer pagurus: Ecological and human health implications. Journal of Food Composition and Analysis, 22, 65-71
11) l.125-127: delete
Reply: Done.
12) l.141: define “dressing percentage”
Reply: We have added the formula of “dressing percentage” in section of Data analysis.
13) l.157-158: I believe that the PCBs and PAHs determined should be at least mentioned in the text
Reply: The PCBs and PAHs are listed in the supplement Table S4-Table S5.
14) l.187: give abbreviations RC, SRC, and EAAI in full
Reply: Done
15) l.195: add “quality” after “nutrition”
Reply: Done
16) Table 2: statistical exponents are missing
Reply: “RC” is measured by comparing the amino acids in a serving of food with an equivalent amount of the model amino acid. SRC is used to assess the nutritional value of proteins. EAAI is the geometric mean of the ratio of the essential amino acid content of proteins to the essential amino acid content of standard proteins (usually egg proteins) and is one of the indicators used to evaluate the nutritional value of food proteins. These indices are calculated based on the average value of the content. The above similar methods were performed in the following references:
[a] Ayodeji Ahmed, A. (2022). Mineral and amino profile of crab (Sudanonaonautes aubryi). Food Chemistry Advances, 1, 100070.
[b] Bi, S., Xue, C., Wen, Y., Li, Z., Liu, H. (2023). Comparative study between triploid and diploid oysters (Crassostrea gigas) on non-volatile and volatile compounds. LWT, 179, 114654.
17) l.211: add “content” after “Fatty acid”
Reply: Done.
18) l.213: change “measured” to “determined”
Reply: Done.
19) l.245: add “content” after “Vitamin and mineral”
Reply: Done.
20) l.246: change “contents” to “content”
Reply: Done.
21) l.255: change “performed” to “calculated
Reply: Done.
22) l.259-260: rewrite in proper English
Reply: The sentence has been revised to “Then, they can be used as an excellent dietary source of vitamin D and B12, which aligns with report on Chlamys farreri”.
23) l.263: delete “with”
Reply: Done.
24) l. 263: sentence does not make sense. Rewrite in proper English
Reply: We have rewritten this sentence “Minerals cannot be synthesized in humans and therefore must be obtained by the diet”.
25) l.264: change “constant” to “macro”
Reply: Done.
26) l.268: what of the authors mean by “unclear regular characteristics” ?
Reply: We deleted this fuzzy words since it doesn’t mean anything.
27) l.269: change “is” to “was”
Reply: Done.
28) Table 3: statistical exponents are missing
Reply: (INQ, Index of Nutrition Quality) = (consumed amount of a nutrient per 1000 kcal/Recommended Dietary Allowance or adequate intake of that nutrient per 1000 kcal). The calculation for INQ is based on the mean value of the content. The similar reports were performed in the following references:
[a] Vahid, F., Hatami, M., Sadeghi, M., Ameri, F., Faghfoori, Z., Davoodi, S. H. (2018). The association between the Index of Nutritional Quality (INQ) and breast cancer and the evaluation of nutrient intake of breast cancer patients: A case-control study. Nutrition, 45, 11-16.
[b] Orkusz, A. (2021). Edible Insects versus Meat-Nutritional Comparison: Knowledge of Their Composition Is the Key to Good Health. Nutrients, 13.
[c] Song, D., Peng, J., Zhao, X., Wu, H., Zheng, G., Zhao, Y., Jiang, Y., Sheng, X., Guo, M., Tan, Z. (2023). Quality and safety profiles of Chlamys farreri cultured in the Shandong peninsula: Analysis of nutritional content, flavor, and hazards. Journal of Food Composition and Analysis, 118, 105193.
29) l.282: change “Taste substances” to “taste contributing compounds”
Reply: Done.
30) l.304: change “presentation” to “attribute”
Reply: Done.
31) l.310, 326: change taste-presenting” to “taste contributing”
Reply: Done.
32) l.31&: give abbreviation EUC in full
Reply: We have given the abbreviation EUC in full.
33) Table 4: statistical exponents are missing
Reply: The taste activity value is the ratio of the concentration of flavor substances in the sample to the corresponding threshold value. The calculation for TAV is based on the mean value of the content. The similar reports were performed in the following references:
[a] Bi, S., Xue, C., Wen, Y., Li, Z., Liu, H. (2023). Comparative study between triploid and diploid oysters (Crassostrea gigas) on non-volatile and volatile compounds. LWT, 179, 114654.
[b] Cong, X., Wang, Q., Sun, C., Yu, F., Chen, L., Sun, Z., Shi, H., Xue, C., Li, Z. (2022). Temperature effects on the nutritional quality in Pacific oysters (Crassostrea gigas) during ultraviolet depuration. Journal of the Science of Food and Agriculture, 102, 1651-1659.
34) l.342: delete “month-old”. Instead, add “monthly” on l.347
Reply: In this study, month-old was used as a variable for this correlation analysis, and the index was not employed and correlated every month.
35) l.359: change “are” to “were”
Reply: Done.
36) l.368: change “taste substances” to “taste contributing compounds”
Reply: Done.
37) 370-371: rewrite in proper English
Reply: We have rewritten these sentences “This study elucidates the characteristic profile in nutritional quality and safety of two shell color bay scallops during their growth. Golden et al. reported the role of aquatic foods in improving human nutrition, especially from pelagic fish and shellfish, which can supply critical nutrients, such as calcium, iron, DHA+EPA, zinc, vitamin B12, and vitamin A. This study can provide a reference to consumers and farmers about their choice of varieties”
Reviewer 2 Report
Comments and Suggestions for Authors
Foods: foods-2707374
Title: "Characteristic Profile in Hazards, Nutritional, and Taste Components During Growth of Argopecten irradians with Different Shell Colors”
These paper is written extensively and clearly. Paper consists of duty parts, which are necessary for structure of scientific work for mentioned journal. The performance of paper is on good level and corresponds to demands, which are generally asked for scientific papers. Text is written on the understanding way of professional language expression. This paper may interest to scientists and other professionals practitioners and consumers.
However, there are some matters which need to be checked and corrected.
- In my opinion, it is not worth repeating the same words in the topic and keywords. It may be better to use other equivalents that will be adequate to the topic presented in the manuscript.
- The quality of Figure 2, Figure 4 and Figure 5 should be improved - they are not legible in their current form.
- The conclusions are insufficient. Whether the purpose of the work was related to practical aspects such as breeding? Please describe the possibilities related to the research conducted in relation to aquaculture or consumer health.
Conclusion: on the basis of above introduced arguments, I evaluate this scientific paper as acceptable after proposed corrections.
Author Response
Thank you very much for taking the time to review this manuscript. Please find the detailed responses below and the corresponding revisions
1)In my opinion, it is not worth repeating the same words in the topic and keywords. It may be better to use other equivalents that will be adequate to the topic presented in the manuscript.
Reply: We have changed “Argopecten irradians” to “Bay scallop”. The similar reports were performed in the following references:
[a] Tang, Y., Feng, L., Jiang, W., Wu, P., Liu, Y., Zhang, L., Kuang, S., Ren, H., Jin, X., Li, S., Mi, H., Zhou, X. (2023). Enhancements in flavor substances, mouthfeel characteristics and collagen synthesis in the muscle of sub-adult grass carp (Ctenopharyngodon Idella): Application of a dietary lysine nutrition strategy. Aquaculture, 565, 739115.
[b] Wang, Q., Sun, C., Chen, L., Shi, H., Xue, C., Li, Z. (2022). Evaluation of microalgae diets on flavor characteristics of Pacific oysters (Crassostrea gigas) during fattening. Food Chemistry, 391, 133191.
2) The quality of Figure 2, Figure 4 and Figure 5 should be improved - they are not legible in their current form.
Reply: Done
3)The conclusions are insufficient. Whether the purpose of the work was related to practical aspects such as breeding? Please describe the possibilities related to the research conducted in relation to aquaculture or consumer health.
Reply: This study elucidates the characteristic profile in nutritional quality and safety of two shell color bay scallops during their growth.
Golden et al. reported the role of aquatic foods in improving human nutrition, especially from pelagic fish and shellfish, which can supply critical nutrients, such as calcium, iron, DHA+EPA, zinc, vitamin B12, and vitamin A. This study can provide a reference to consumers and farmers about their choice of varieties.
[a] Golden, C. D., Koehn, J. Z., Shepon, A., et al. (2021). Aquatic foods to nourish nations. Nature, 598, 315-320.
Reviewer 3 Report
Comments and Suggestions for Authors
REVISION
The manuscript entitled “Characteristic Profile in Hazards, Nutritional, and Taste Components During Growth of Argopecten irradians with Different Shell Colors” aims to analyze the seawater environmental parameters, microbial and chemical hazards, biometric measurements, nutritional components and flavor substance of cultured A. irradians during its growth in Rongcheng Ailian Bay, to provide both valuable insights applicable to A. irradians aquaculture, production methods and innovative processing techniques, and a valuable reference for the culture of scallops with a golden or purple shells, contributing to the improvement of new nutrient-rich and flavorful scallop varieties.
The purpose of the following study fits with that of the journal Foods (MDPI).
The study is very interesting, well written and structured. Many different analyses have been carried out on the samples of interest, which greatly enhances the scientific value of this study. The introduction first explains the importance of scallops in the Chinese aquaculture industry. This is followed by an explanation of the times at which these matrices are caged, reared, harvested, and sold. It also highlights some of the nutritional characteristics of scallops, such as their important protein content.
Minor revisions are required.
2. Materials and Methods
2.4. Experiment method
-Page 3, lines 104-106: I understand that you have used previously optimized methods. However, I would recommend that you at least mention the preparation methods used for each individual analysis.
3. Results and Discussion
3.3. Microbial and chemical hazards
-Page 5, line 155: I would like to refer to the legislation which lists standard limits for lead, cadmium, arsenic, mercury.
3.4. Proximate composition
-Page 5, line 173: Is there any explanation for this trend?
3.6. Fatty acid
-Page 7, line 227: This relationship is also important because, if unbalanced, it can lead to chronic diseases due to competition between n-3 and n-6 PUFAs for the same enzymes. I suggest you read the following scientific article: Nava, V.; Turco, V.L.; Licata, P.; Panayotova, V.; Peycheva, K.; Fazio, F.; Rando, R.; Di Bella, G.; Potortì, A.G. Determination of Fatty Acid Profile in Processed Fish and Shellfish Foods. Foods 2023, 12, 2631. https://doi.org/10.3390/ foods12132631
3.7. Vitamin and mineral
-Page 8, lines 262-263: I recommend rephrasing this sentence as it is unclear.
-Page 8, lines 264-266: Do you have a reference that show the use of this term? Otherwise, I would recommend using the term 'macro elements'.
-Page 9, lines 275-276: Is there an explanation for these trends?
Author Response
Thank you very much for taking the time to review this manuscript. Please find the detailed responses below and the corresponding revisions.
- Materials and Methods
2.4. Experiment method.
-Page 3, lines 104-106: I understand that you have used previously optimized methods. However, I would recommend that you at least mention the preparation methods used for each individual analysis.
Reply: The analytical methods are the classic methods of our group. Details of the analysis methods have been listed in the supplemental section.
- Results and Discussion
3.3. Microbial and chemical hazards
-Page 5, line 155: I would like to refer to the legislation which lists standard limits for lead, cadmium, arsenic, mercury.
Reply: Done.
- 4. Proximate composition
-Page 5, line 173: Is there any explanation for this trend?
Reply: This indicates that most of the energy reserves were stored in A. irradians, which prepares its growth and gametogenesis.
[a] Kang, H. Y., Lee, Y., Lee, W., Kim, H. C., Kang, C. (2019). Gross biochemical and isotopic analyses of nutrition-allocation strategies for somatic growth and reproduction in the bay scallop Argopecten irradians newly introduced into Korean waters. Aquaculture, 503, 156-166.
- 6. Fatty acid.
-Page 7, line 227: This relationship is also important because, if unbalanced, it can lead to chronic diseases due to competition between n-3 and n-6 PUFAs for the same enzymes. I suggest you read the following scientific article: Nava, V.; Turco, V.L.; Licata, P.; Panayotova, V.; Peycheva, K.; Fazio, F.; Rando, R.; Di Bella, G.; Potortì, A.G. Determination of Fatty Acid Profile in Processed Fish and Shellfish Foods. Foods 2023, 12, 2631. https://doi.org/10.3390/ foods12132631
Reply: The sentences have been rephrased “EPA and DHA are important components of n-3 PUFAs in the bay scallops. Existing study shows that if n-3/n-6 is unbalanced, it can lead to chronic diseases due to competition between n-3 and n-6 PUFAs for the same enzymes. In this study, the ratios of n−3/n−6 of bay scallops were all greater than 8, suggesting that bay scallops can provide good n−3 unsaturated fatty acids and are suitable for human consumption.”
- Vitamin and mineral.
Page 8, lines 262-263: I recommend rephrasing this sentence as it is unclear.
Reply: We have rewritten this sentence “Minerals cannot be synthesized in humans and therefore must be obtained by the diet”.
- -Page 8, lines 264-266: Do you have a reference that show the use of this term? Otherwise, I would recommend using the term “macro elements”.
Reply: The “constant elements” has been changed to “macro elements” in the revision.
- -Page 9, lines 275-276: Is there an explanation for these trends?
Reply: “similar results can be found in oysters, which may be related to differences in growth metabolism across shell colors.”
[a] Zhu, Y., Li, Q., Yu, H., Kong, L. (2018). Biochemical Composition and Nutritional Value of Different Shell Color Strains of Pacific Oyster Crassostrea gigas. Journal of Ocean University of China, 17, 897-904.
Reviewer 4 Report
Comments and Suggestions for Authors

Comments on the Quality of English Language
In general the quality of English is good
Author Response
1)The Introduction could use a bit more of a background on the topic related to scallops.
Reply: Literature on the nutritional aspects of bay scallops is limited and relevant literature has been cited in the introduction.
2)Page 2- Line 91-93. ………, salinity, pH, dissolved oxygen, chlorophyll a, and nutrients were taken.
Reply: Done
3) Page 2- Line 100. How many animals were sampled and on how many animals biochemical analyzes were conducted?
Reply: We have added “4 kg samples from each site were kept with ice and transported to the laboratory. Shell height, length, and width, total weight, and tissue weights were measured. All the samples were divided into two parts: one for determining microbial, chemical hazard, moisture, and taste contributing compounds, and the other was freeze-dried and stored at −80℃ for subsequent fat, protein, hydrolyzed amino acid, fatty acid, vitamin, and mineral analysis” to “Sample pretreatment”
4) Page 4- Line 130. Specify the period to make the data more immediate for the reader …water temperature and salinity were in the range 17.13 (October) –23.50℃ (September) and 23.20 (September) –26.70 (August), respectively
Reply: Done
5) Table S1- In the table S1, standardizes the salinity data with the others by adding the second digit Dissolved oxygen content in the aquaculture waters varied from 6.31 mg/L in August to 9.79 mg/L in October…..and so on.
Reply: Done
6) Page 4- Line 139-143. In my opinion, it is not correct to present the data in this way. For example, as regard the shell length, 35.24 mm refers to golden shell individuals, while 68.33 mm to purple shell individuals…, but they cannot be understood from how they are written
Reply: The content ranges for each shell color scallop have been added, similar expressions can be found in the following references.
[a] Barrento, S., Marques, A., Teixeira, B., Anacleto, P., Vaz-Pires, P., Nunes, M. L. (2009). Effect of Season on the Chemical Composition and Nutritional Quality of the Edible Crab Cancer Pagurus. Journal of Agricultural and Food Chemistry, 57, 10814-10824.
7) Page 4- Line 148. Tables S2–7 and Figure 2 reveal the microbial and chemical hazard results……… write better, because the first tables refer to chemical hazard and the last to microbial results, while in the text you presented microbial results first and so on with the other results. In this way it is confused.
Reply: Done
8) Table S8 shows data in g/100 g, whereas in the figure 3 the same data are reported as mg/100 g….??????
Reply: The uncorrected units in Figure 3 has been revised.
9) Page 6- Line 190-191. …..with a range of 42.14–57.01 g/100 g from August to October……Where in the table????? You have to correct.
Reply: We have revised “42.14–57.01 g/100 g” to” 42.10–56.99 g/100g”.
10) Page 6- Line 197-198. The essential amino acid content in A. irradians growth was 11.31–12.91 g/100 g…. Where in the table????? You have to correct.
Reply: All data has been checked and revised thoroughly, as a result it has been confirmed correct.
11) Page 6- Line 203. ……….. during growth was 44.24–48.07 (Table 2).
Reply: Done
12) Page 6- Line 207-208. The SRC values were 92.66–96.03….check the data.
Reply: We have corrected the Page 6- Line 207-208, and changed “92.66–96.03” to” 92.33–98.21”
13) Figure 4. To make the two graphs uniform, check the graph on the right and move the fatty acid C18:3n3. Moreover, in the caption Figure 4. The fatty acid content ……add (mg/100g)
Reply: The position of fatty acid C18:3n3 in Figure 4 has been revised. The fatty acid content could be found in the original horizontal coordinate.
14) Page 7- Line 216-224 . The SFAs of A. irradians accounted for 26.33%–34.94% of the total fatty acids… MUFAs represented 16.30%–21.97%,…… PUFAs, which accounted for 46.24%–56.09%..... Since the data of the percentage content of SFA, MUFA and PUFA are not reported in the table and figure, but only the range is reported in the text without reference to the two shell color strains and months, in my opinion you should write that there are no statistical differences between two shell color strains and months.
Reply: The fatty acid composition has been added as Table S11 in the supplement.
15) Page 7- Line 224-225. The n-3 series of PUFA of EPA and DHA is important in the human body. You should write better.
Reply: We have rewritten the sentences “EPA and DHA are important components of n-3 PUFAs in the bay scallops. Existing study shows that if n-3/n-6 is unbalanced, it can lead to chronic diseases due to competition between n-3 and n-6 PUFAs for the same enzymes. In this study, the ratios of n−3/n−6 of bay scallops were all greater than 8, suggesting that bay scallops can provide good n−3 unsaturated fatty acids and are suitable for human consumption”.
16) Page 7- Line 233-238. Comparisons with literature data on other species and genera are also missing, which are important because they highlight the interesting results in this regard even more.
Reply: The literature about oysters has been added in the revision.
17) Page 10- Line 314. …..ranging from 3.86 to 17.25 mg/100 g and 5.70 to 34.47 mg/100 g, respectively. Check the data
Reply: The data has been checked thoroughly, and been revised in the manuscript.
1)Page 2- Line 91-93. ………, salinity, pH, dissolved oxygen, chlorophyll a, and nutrients were taken.
Reply: Done
2) Page 2- Line 100. How many animals were sampled and on how many animals biochemical analyzes were conducted?
Reply: We have added “4 kg samples from each site were kept with ice and transported to the laboratory. Shell height, length, and width, total weight, and tissue weights were measured. All the samples were divided into two parts: one for determining microbial, chemical hazard, moisture, and taste contributing compounds, and the other was freeze-dried and stored at −80℃ for subsequent fat, protein, hydrolyzed amino acid, fatty acid, vitamin, and mineral analysis” to “Sample pretreatment”
3) Page 4- Line 130. Specify the period to make the data more immediate for the reader …water temperature and salinity were in the range 17.13 (October) –23.50℃ (September) and 23.20 (September) –26.70 (August), respectively
Reply: Done
4) Table S1- In the table S1, standardizes the salinity data with the others by adding the second digit Dissolved oxygen content in the aquaculture waters varied from 6.31 mg/L in August to 9.79 mg/L in October…..and so on.
Reply: Done
5) Page 4- Line 139-143. In my opinion, it is not correct to present the data in this way. For example, as regard the shell length, 35.24 mm refers to golden shell individuals, while 68.33 mm to purple shell individuals…, but they cannot be understood from how they are written
Reply: The content ranges for each shell color scallop have been added, similar expressions can be found in the following references.
[a] Barrento, S., Marques, A., Teixeira, B., Anacleto, P., Vaz-Pires, P., Nunes, M. L. (2009). Effect of Season on the Chemical Composition and Nutritional Quality of the Edible Crab Cancer Pagurus. Journal of Agricultural and Food Chemistry, 57, 10814-10824.
6) Page 4- Line 148. Tables S2–7 and Figure 2 reveal the microbial and chemical hazard results……… write better, because the first tables refer to chemical hazard and the last to microbial results, while in the text you presented microbial results first and so on with the other results. In this way it is confused.
Reply: Done
7) Table S8 shows data in g/100 g, whereas in the figure 3 the same data are reported as mg/100 g….??????
Reply: The uncorrected units in Figure 3 has been revised.
8) Page 6- Line 190-191. …..with a range of 42.14–57.01 g/100 g from August to October……Where in the table????? You have to correct.
Reply: We have revised “42.14–57.01 g/100 g” to” 42.10–56.99 g/100g”.
9) Page 6- Line 197-198. The essential amino acid content in A. irradians growth was 11.31–12.91 g/100 g…. Where in the table????? You have to correct.
Reply: All data has been checked and revised thoroughly, as a result it has been confirmed correct.
10) Page 6- Line 203. ……….. during growth was 44.24–48.07 (Table 2).
Reply: Done
11) Page 6- Line 207-208. The SRC values were 92.66–96.03….check the data.
Reply: We have corrected the Page 6- Line 207-208, and changed “92.66–96.03” to” 92.33–98.21”
12) Figure 4. To make the two graphs uniform, check the graph on the right and move the fatty acid C18:3n3. Moreover, in the caption Figure 4. The fatty acid content ……add (mg/100g)
Reply: The position of fatty acid C18:3n3 in Figure 4 has been revised. The fatty acid content could be found in the original horizontal coordinate.
13) Page 7- Line 216-224 . The SFAs of A. irradians accounted for 26.33%–34.94% of the total fatty acids… MUFAs represented 16.30%–21.97%,…… PUFAs, which accounted for 46.24%–56.09%..... Since the data of the percentage content of SFA, MUFA and PUFA are not reported in the table and figure, but only the range is reported in the text without reference to the two shell color strains and months, in my opinion you should write that there are no statistical differences between two shell color strains and months.
Reply: The fatty acid composition has been added as Table S11 in the supplement.
14) Page 7- Line 224-225. The n-3 series of PUFA of EPA and DHA is important in the human body. You should write better.
Reply: We have rewritten the sentences “EPA and DHA are important components of n-3 PUFAs in the bay scallops. Existing study shows that if n-3/n-6 is unbalanced, it can lead to chronic diseases due to competition between n-3 and n-6 PUFAs for the same enzymes. In this study, the ratios of n−3/n−6 of bay scallops were all greater than 8, suggesting that bay scallops can provide good n−3 unsaturated fatty acids and are suitable for human consumption”.
15) Page 7- Line 233-238. Comparisons with literature data on other species and genera are also missing, which are important because they highlight the interesting results in this regard even more.
Reply: The literature about oysters has been added in the revision.
16) Page 10- Line 314. …..ranging from 3.86 to 17.25 mg/100 g and 5.70 to 34.47 mg/100 g, respectively. Check the data
Reply: The data has been checked thoroughly, and been revised in the manuscript.
Round 2
Reviewer 1 Report
Comments and Suggestions for Authors
Τhe following corrections are recommended in the revised text:
l.43: change 'about' to 'on'
l.46: change 'nutrition' to 'nutritional value'
l.64: delete 'types of'
l.76, 77:change 'chromatography' to 'chromatograph'
l.98: change 'with' to 'in'
l.130,131:give temperature figure with one decimal point
l.141,142: give length figure with one decimal point if length was measured with a ruler
l.282: change 'by' to 'through'
Comments on the Quality of English Language
Author Response
1) l.43: change 'about' to 'on'
Reply: Done
2) l.46: change 'nutrition' to 'nutritional value'
Reply: Done
3) l.64: delete 'types of'
Reply: Done
4) l.76, 77: change 'chromatography' to 'chromatograph'
Reply: Done
5) l.98: change 'with' to 'in'
Reply: Done
6) l.130,131: give temperature figure with one decimal point
Reply: Done
7) l.141,142: give length figure with one decimal point if length was measured with a ruler
Reply: Done
8) l.282: change 'by' to 'through'
Reply: Done
Reviewer 4 Report
Comments and Suggestions for Authors
the paper is significantly improved
Author Response
Thank you very much for your comments and suggestions!